# VIMPNN: PHYSICS INFORMED DNN TO ESTIMATE POTENTIAL ENERGIES OF UNSTABLE SYSTEMS

## ABSTRACT

Simulation of molecular and crystalline systems enables insight into interesting chemical properties that benefit processes ranging from drug discovery to material synthesis. However, these simulations can be computationally expensive and time consuming despite the approximations through Density Functional Theory (DFT). We propose the Valence Interaction Message Passing Neural Network (VIMPNN) to approximate DFT's potential energy calculations. VIMPNN integrates prior physics knowledge, namely the information exchange between atoms being driven and modulated by different interatomic bonds, and the relevance of specific physical properties to the problem of estimating more accurate energies. Furthermore, while many previous machine learning methods consider only stable systems, our proposed method is demonstrated on unstable systems at different interatomic distances. VIMPNN predictions can be used to determine the stable locations for atoms, a necessary step for the future simulation of crystal growth. Our method is extensively evaluated on an augmented version of the QM9 dataset that includes unstable molecules, as well as 2 new datasets of periodic and growing crystals. VIMPNN has comparable accuracy with DFT, while allowing for 5 orders of magnitude in computational speed up. It also produces more accurate and informative potential energy curves than Message Passing Neural Network.

## 1 INTRODUCTION

Chemical simulations have many useful industrial applications ranging from drug discovery to the production of materials for daily use (Schütt et al., 2017). Simulating crystal systems in particular provides useful properties such as surface absorption, chemical reactions, and surface magnetism (Bilek & Skála, 1978). Quantum Mechanical (QM) simulations can be used for the calculation of potential energies based upon the interaction of atoms. By simulating a chemical system at different interatomic distances, it is possible to determine a stable configuration where the atomic interaction is at an equilibrium (minimum potential energy). While QM simulations can be less time consuming than physical experimentation, they typically require large amounts of computing resources and do not scale well to larger system sizes (Jiang et al., 2003; Erba et al., 2017).

To address this difficultly in determining the potential energy, the Kohn-Sham Density Functional Theory (DFT) may be used to simplify the calculations by considering the electronic density in place of individual electrons. DFT has proved useful in QM due to its good trade-off between speed of computation and chemical accuracy (Cohen et al., 2012). However, as DFT calculations are proportional to the number of interacting electrons (Lanyon et al., 2010), computation for large systems remains intractable. As many interesting and realistic systems are formed from a large number of atoms, a computationally efficient and accurate method for chemical property estimation that scales well to these large systems would prove significantly useful in practical applications (Gomes et al., 2008). Moreover, several simulations are often necessary to identify stable configurations away from local minima. Thus, fast approximation methods are desirable in such scenarios.

Machine learning (ML) was proposed to efficiently classify (stable) molecules and estimate chemical properties by reproducing the results of DFT, with recent advancements through a Message Passing Neural Network (MPNN) (Gilmer et al., 2017). However, unstable molecules and crystals are often not considered, although this would allow discovering new stable configurations.

In this article we demonstrate the applicability of ML for the energy prediction of *unstable* molecular and crystalline systems, where the atoms are at positions which are not at the DFT calculated minimum of potential energy, in the following referred to as either "unstable" or "out-of-equilibrium". These energies are determined at the electronic ground-state at given positions of atoms for static systems. We propose a new DNN framework, the VIMPNN (Valence Interaction Message Passing Neural Network), that considers the different interatomic interactions driven by different valences. It produces comparable accuracy to that of DFT while also improving the computation time by 5 orders of magnitude. We demonstrate that our method also produces more accurate energy estimations than that of MPNN, while being 30% slower at 0.007 seconds per energy estimate, which is marginal compared to the gain from DFT and may be improved with some obvious code optimisation.

ML in quantum chemistry requires the availability of large datasets of chemical structures and their (measured or computed) chemical properties. Online repositories such as Quantum Machine[1] store a collection of datasets resulting from various QM simulations. QM9 (Blum & Reymond, 2009; Montavon et al., 2013) is one such dataset containing 134K molecules using carbon, hydrogen, oxygen, nitrogen, and fluorine atoms. However, as all molecules in this dataset are at their stable configuration, it is not suitable for designing and testing accurate energy prediction methods for unstable systems. We address this shortcoming by augmenting the QM9 dataset with out-of-equilibrium configurations for 10K of its molecules at a [90%, 150%] range of interatomic distances. In addition, we create two datasets composed of infinite-size periodic crystals and finite-size growing crystals of aluminium (Al) and copper (Cu) atoms.

In summary, the contributions of this work are:

- A new VIMPNN model that accounts for the physics of atomic bonds within a molecule or crystal to improve the estimation of chemical properties. This model also introduces auxiliary chemical property estimations to help learn descriptors that are closer to the physics of the problem.
- A new use of ML for potential energy estimation of unstable molecules and crystals.
- The public release of new periodic crystal and growing crystal datasets and an augmented QM9 dataset.

The remainder of this article is organised as follows: Section 2 discusses previous works on molecular properties estimation by ML. We describe the creation of 3 new datasets in Section 3. In Section 4 we introduce our new physics informed DNN for accurate energy estimation, which we evaluate in Section 5. We conclude in Section 6.

## 2 RELATED WORK

Several ML methods have been used for estimating a variety of chemical properties from different representations of chemical systems. Li et al. (2015) constructed a covariance matrix from Euclidean distances of atoms within molecules, then use Bayesian regression to estimate interatomic forces. Wang et al. (2013) used kernel ridge regression on the numbers of coarse *'building blocks'* (such as $CH_2$) in molecules to estimate many chemical properties such as electron affinity and atomization energy. Recent research by Shi et al. (2019) used strain information and a Fourier transform-based representation of crystal lattice for a feed-forward neural network (NN) to estimate an electronic bandgap structure in silicon. The bandgap refers to the energy difference between the valence and conduction bands in insulators and semiconductors, and therefore the energy required to transition between stable excitation states. Silicon crystal in a known equilibrium state is strained in the range of -10% to 10% in each strain component (such as thermal properties) to search the bandgap space for another stable state, for example transforming silicon from a semiconductor to a metal. Therefore, this work uses the NN to describe the bandgap as a function of the strain tensors. In contrast, our method infers the effect of spatial deformations on a system's potential energy.

Some approaches focused on the prediction of stable geometries for chemical systems. Timoshenko et al. (2018) used experimental spectra data as input for an NN to estimate the probability distribution of the stable bonding distance between pairs of atoms. They considered different combinations of

---

[1]http://quantum-machine.org/

Pd and Au atoms (i.e. an NN for Au-Au bonds, another for Pd-Au). When considering crystals made of two different atom types, Takahashi & Takahashi (2019) used 8 quantities associated with each atom type (such as atomic radii, electronegativity, and number of atoms) to perform a random forest classification on how the resulting stable crystal structure would form among a set of 492 different possible structures. The predicted lattice type requires further optimisation using DFT to find its exact size/scaling. However, this method may also predict metastable configurations (i.e. local minima). Whereas the aforementioned methods attempted to directly obtain the stable distance between atoms, we estimate potential energies for various geometries with the aim to minimise them into stable configurations. While this approach may require more training data and more complex modelling to handle arbitrary configurations, we argue that it has a stronger generalisation potential. It also has the added benefit of allowing the (future) observation of progressive crystal growth.

Previous works on estimating the potential energy of chemical systems include that of Rupp et al. (2012) who used a feed-forward NN on the Coulomb matrix (a pairwise matrix that describes the low-level electrostatic interaction between atoms) of molecules. However, predictions on this matrix are sensitive to permutations of atoms which can result in different property values being estimated for a same crystal. Montavon et al. (2012) provided an invariant solution by training the NN on a set of randomly permuted matrices. To show the benefit of their method, they train different ML models on the QM7 dataset and improved on the accuracy of previous approaches by a factor of 3. A recent advancement by Gilmer et al. (2017) provided a more versatile representation for molecules as a chemical graph where nodes describe atoms and edges encode the distances and bond type between them. Their Message Passing Neural Network (MPNN) simulates the atomic interactions through the passing of messages between the nodes. So far, this work has only been evaluated on stable configurations of molecules from the QM9 dataset. We further evaluate MPNN on unstable molecules and crystals. Furthermore, we extend on their approach to allow for more accurate predictions by better accounting for the physics of atomic interactions.

## 3    DATASETS

Many previous ML approaches and associated datasets only consider the case of molecules and crystals at their stable configuration, therefore learning the interaction of atoms only at an equilibrium state. The estimation of potential energies for unstable systems needs to be supported by a dataset of unstable systems. We created 3 such datasets to train and benchmark VIMPNN. The first is an augmented QM9 dataset (Section 3.1) to investigate the complexity of atomic interactions from a variety of atom types in small but diverse molecules. In the second dataset, infinite-size crystals (Section 3.2) allow learning regular bonding patterns that arise in periodic structures. Thirdly, a dataset containing growing crystals of an increasing (finite) size and complexity enables experimenting with large scale interactions (Section 3.3).

### 3.1    AUGMENTED QM9 DATASET

An augmented QM9 dataset is created by taking the first 10,000 molecules of QM9 and modifying the interatomic distances at 10 regular intervals between 90 and 150% of the original stable configuration. This dataset therefore contains 100K different systems. At each interval, the potential energy is calculated using DFT[2] to serve as the target energy for our supervised learning task.

### 3.2    INFINITE CRYSTAL DATASET

Learning to estimate the potential energy for an infinitely sized crystal might benefit from the regular pattern in the lattice structure, possibly reducing the complexity of modelling for the ML algorithm. A dataset of infinite-sized crystals, although not necessarily relevant from a physical point of view, is therefore an interesting case study for the design and training of ML methods. We use the simple Face-Centred Cubic (fcc) Bravais lattice illustrated in Fig. 1 left for both Al and Cu atoms. We iteratively compress/dilate the cell size (thus, the interatomic distances) and compute the potential energies using DFT at 20,000 uniform intervals from 90-150% of the stable configuration. The change of atomic distances are performed isometrically (i.e. the same change in all spatial axes).

---

[2]We used CP2K (`https://www.cp2k.org/`) to calculate the potential energies with DFT.

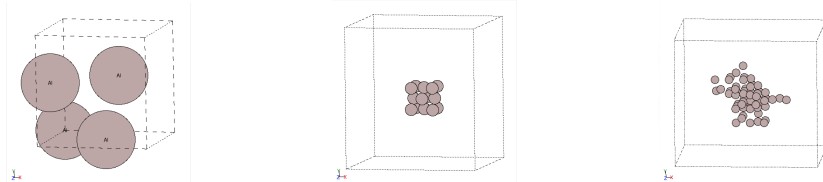

Figure 1: Crystal structures for the Infinite Crystal (left) and Crystal Growth (centre: seed, right: growing) datasets.

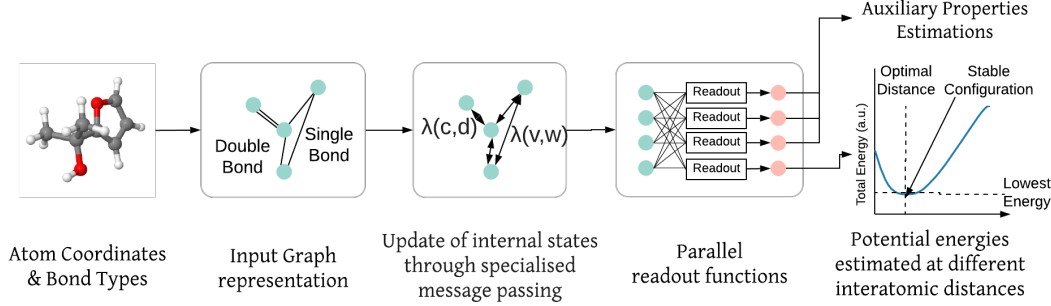

Figure 2: Outline of VIMPNN: physics-informed DNN for estimating potential energies from a graph representation of chemical systems.

## 3.3 CRYSTAL GROWTH DATASET

In contrast to the regular lattice pattern created by an infinite crystal, crystal growth allows for learning over more complex atomic interactions in non-regular systems. Starting from a basic fcc crystal seed of 14 atoms (Fig. 1 centre), new crystalline systems are generated by iteratively placing atoms at a random location on the surface of the growing crystal following its lattice pattern (Fig. 1 right), with sizes ranging from 15 to 114 atoms. Such a dataset enables evaluation of an ML method's ability to learn how each atom contributes to the final potential energy.

We generate two subsets of this dataset: one that includes compression/dilation of the interatomic distances at 10 regular intervals between 90 to 150% of the stable distance, referred to as the Unstable Crystal Growth subset, and another that contains only stable configurations, referred to as the Stable Crystal Growth subset. The latter aims to evaluate an ML model's capability to estimate energy values at varying system sizes (i.e. number of atoms) without the added complexity of learning energy as a function of distance. Both subsets contain Al and Cu crystalline systems. For the Stable subset, we use 20 random seeds for each atom type, thus creating 40 varied crystal growths and 4,000 crystalline systems. For the Unstable subset, we select 5 of these random seeds for each atom type to consider 1,000 basic configurations, each being compressed/dilated for a total of 10,000 crystalline systems.

## 4 METHODOLOGY

VIMPNN estimates the potential energies for molecular or crystaline systems through learning the energy as a function of the geometry of bonded atoms. By querying VIMPNN at various interatomic distances, we may find a stable configuration (Fig. 2). VIMPNN extends MPNN (Gilmer et al., 2017) that simulates atomic interactions through the passing of messages between pairs of atoms (Section 4.1). However, VIMPNN further accounts for the physics of the system through an architecture that passes information according to the different forms of bonding between atoms (Section 4.2), and through reinforcing the physical relevance of the analysis and learnt features using auxiliary physical properties (Section 4.3). A representation of the VIMPNN model is depicted in Fig. 2.

### 4.1 PREREQUISITE: MPNN

MPNN learns to predict potential energy from an undirected graph $\mathcal{G}$, where atoms are nodes and edges $e_{vw}$ describe the relation between two nodes $v$ and $w$ (e.g. their distance). MPNN applies three functions for a fixed number of iterations (recommended between 3 and 8): 1) the Message function ($M_t$ in Eq. 1, implemented with a perceptron) creates a *message packet* that symbolises the action of a neighbouring atom on node $v$. This action does not replicate the physics of atomic interactions happening in real chemical systems, but it enforces the NN model to process information in a physics-inspired way. A final *message* vector $m_v^t$ for node $v$ is created in Eq. 1 by combining the actions of all neighbours. 2) An update function ($U_t$ in Eq. 2) updates the hidden state $h_v^t$ of node $v$ using its previous hidden state $h_v^{t-1}$ and message $m_v^t$. It accounts for the effect of atomic interactions on the (information) properties of the node's atom. It is implemented by a Gated Recurrent Unit (GRU). 3) The Readout function ($R$ in Eq. 3, implemented with a perceptron) uses the set of hidden state of all nodes at all timesteps to estimate the system's potential energy. Thus, it exploits the properties of atoms and of their exchanges of information, to learn to estimate the energy through minimising a mean-squared error loss.

$$m_v^{t+1} = \sum_{w \in N(v)} M_t\left(h_w^t, e_{vw}\right) \tag{1}$$

$$h_v^{t+1} = U_t\left(h_v^t, m_v^{t+1}\right) \tag{2}$$

$$y_{energy} = R\left(\left\{h_v^T | v \in \mathcal{G}\right\}\right) \tag{3}$$

Full details on the implementation of these three functions can be found in the original paper (see Gilmer et al. (2017)). Next, we detail how VIMPNN adapts these functions to further incorporate physics principles in order to better handle out-of-equilibrium systems.

### 4.2 ACCOUNTING FOR BOND TYPES

The type of bonds between different atoms are an important factor when considering their interaction and its contribution to the energy of the system. Chemical bonds between atoms can be characterised by the number of valence electrons exchanged in the process of bonding. For example, ionic bonding requires one donor electron and one acceptor, while in situations where the bonding atoms have similar electronegativity, some electrons would be shared between the bonding atoms resulting in a covalent bond. Moreover, most atomic systems have minimum potential energy (stable bond) at an optimal atom separation (Feinberg & Ruedenberg, 1971). Therefore, the potential energy of a system may be better estimated by accounting for the contribution of different bond types. Gilmer et al. (2017) acknowledged this fact and introduced bond type information as a feature in the edges $e_{vw}$ of the undirected graph representing the chemical system. This information was used together with interatomic distance by the message function $M_t$ in Eq. 1 to estimate the influence of neighbouring atoms on a node. Thus, messages exchanged by the NN nodes are impacted by the bond type. Gilmer et al. (2017) demonstrated this approach results in better energy estimates over using distance alone.

These encouraging results inspired us to take this physics integration principle further and to modify the design of information exchange within the NN to better follow the physics of atomic interactions. We propose to introduce *different specialised communication channels based on bond type*, with a direct effect on the update of nodes (Eq. 2). This is supported by separate messages $m_v|_{BT}$ for each bond type, computed as in Eq. 1 to combine all message packets of a same bond type. Bond type is quickly predetermined based on the atoms' valency and electronegativity using the RDKit software[3]. In practice, the principle of having different messaging channels and node update based on bond type may be implemented in various ways. We experimented with: a) combining the messages $m_v|_{BT}$ as a weighted sum, where weights are learnt by the NN to control the influence of each $m_v|_{BT}$:

$$m_v = \sum_{BT} m_v|_{BT} \cdot \lambda_{BT} \, , \tag{4}$$

where $\lambda_{BT}$ is the weight for bond type $BT$, and $t$ is omitted for readability. We tried with $\lambda_{BT}$ being i) a vector of same size as the message (with an element-wise product in Eq. 4), and ii) a simple scalar. The final combined message $m_v$ is provided to the GRU of Eq. 2 to compute a node

---

[3]https://www.rdkit.org/

update where different bond types contribute differently. Another possibility is: b) concatenating the messages $m_v|_{BT}$ before providing them to the GRU of Eq. 2. We experimented with i) letting the GRU handle the concatenated $m_v|_{BT}$ freely, and with ii) imposing a similar handling of the different $m_v|_{BT}$ through using the same GRU parameters, duplicated for each $m_v|_{BT}$ with a simple (learnt) scaling to allow different weightings of the $m_v|_{BT}$ messages in the node update. Finally, we also experimented with: c) having one GRU for each bond type, handling its corresponding message $m_v|_{BT}$. The updates of all GRUs were summed to compute the final nodes' update:

$$h_v = \sum_{BT} U_{BT}(h_v, m_v|_{BT}) ,\qquad(5)$$

where $U_{BT}$ is the update function implemented by the GRU associated with bond type $BT$, and $t$ being again omitted for readability. As demonstrated in the Appendix (Table 3), we found that the best results were obtained in the case a.ii) with combining all messages $m_v|_{BT}$ in a simple way while letting the NN weight the influence of each bond type specialised message $m_v|_{BT}$.

### 4.3 ESTIMATING AUXILIARY PHYSICAL PROPERTIES

We hypothesise that, by encouraging the VIMPNN's hidden states to relate more to basic physical properties, we may obtain a more accurate energy estimator that generalises better to new systems. In addition, as the bond type is characterised by the valence property of atoms involved in the bonding process, a more physically relevant hidden state may better support the differentiation of messages and node updates per bond type (Section 4.2). In practice, we encourage this greater relationship with physical parameters through the estimation of auxiliary properties through parallel readout functions similar to Eq. 3. We present this strategy as a second and easy way of integrating knowledge (of task-related physical properties) into the model. We experimented with: 1) the number of atoms of each type present in the system, 2) the number of orbitals associated with each atom type – a property that is directly relevant to the determination of bond type –, and 3) a probability distribution for the scaling to the stable distance, estimated as a Gaussian function. For each auxiliary estimation, a mean-squared error loss term (weighted by an $\alpha = 0.3$ hyperparameter) is minimised during training.

## 5 RESULTS AND DISCUSSION

In this section, the performance impact for each of the physics integration strategies introduced in Section 4 are first measured in turn on our augmented QM9 dataset (Section 5.1). The VIMPNN model, that includes all augmentations, is further evaluated on all datasets to demonstrate its versatility on different scenarios (Section 3), using MPNN as a baseline for comparison (Section 5.2). In particular, we test the generalisation ability of VIMPNN by training it on crystalline systems of up to 25 atoms to predict the energies of systems of up to 75 atoms (Section 5.3). We finally explore the physics relevance and interpretability of VIMPNN by visualising its internal states (Section 5.4).

In all experiments, the dataset is split into training, testing, and validation sets in proportions appropriate for the complexity of the data. The Augmented QM9 training set consists of all data augmentations, i.e. 10 compression/dilation operations, for 80% of the QM9 molecules present in the dataset, so 8,000 molecules, totalling 80,000 training samples. Following the same rules, the testing and validation sets each contain 10% of the dataset, so 10,000 samples. As the Infinite Crystal dataset has 2 different crystal types (Cu and Al crystals), 8 data points are uniformly sampled from their respective range of compression/dilation for training. The rest of data is split evenly between validation and test. For the Crystal Growth dataset, we use a 60/20/20 split with the same procedure.

### 5.1 PHYSICS INTEGRATION PERFORMANCES

We first evaluate the effectiveness of the physics integration strategies listed in Section 4 by comparing them, in turn, against the baseline of no physics integration (i.e. $e_{vw}$ only contains distance and no bond type information) and MPNN (i.e. bond type information used in $e_{vw}$). We report, in Table 1, average and standard deviation metrics for the mean-absolute error (MAE), the mean-squared error (MSE), and the relative error (RE $= \frac{|\hat{y}-y|}{|y|}$) between the true $y$ and predicted $\hat{y}$ energy values.

All proposed methods have a positive effect on energy estimation, with the introduction of bond type information (rows 2-3) having a particularly positive impact. This confirms Gilmer et al. (2017)'s

Table 1: Evaluation of the impact of each physics integration strategy on the accuracy of energy estimation on the Augmented QM9 dataset. Results are presented in the format: mean (std).

| | Strategy | MAE | MSE | RE |
|---|---|---|---|---|
| | No bond type (BT) information | 0.220 (1.169) | 1.416 (17.913) | 0.0027 (0.013) |
| | MPNN (BT specialised messages) | 0.091 (0.476) | 0.235 (6.679) | 0.0012 (0.005) |
| | BT specialised node updates (case a.ii) | 0.067 (0.141) | 0.024 (0.406) | 0.0009 (0.002) |
| | # atoms | 0.171 (0.986) | 1.002 (13.940) | 0.0021 (0.011) |
| Auxiliary estimates of | # orbitals | 0.195 (0.545) | 0.335 (7.223) | 0.0025 (0.006) |
| | BT distance scaling | 0.119 (0.698) | 0.501 (9.548) | 0.0015 (0.008) |

Table 2: Evaluation of MPNN and VIMPNN on all proposed datasets.

| Model | Dataset | MAE | MSE | RE | DSC |
|---|---|---|---|---|---|
| MPNN | Augmented QM9 | 0.091 (0.476) | 0.235 (6.679) | 0.0012 (0.005) | 0.039 (0.056) |
| | Infinite Crystal | **0.034** (0.032) | **0.002** (0.005) | **0.0012** (0.002) | 0.201 (0.211) |
| | Stable CG | 2.906 (4.200) | 26.069 (71.750) | 0.0047 (0.005) | - |
| | Unstable CG | 3.447 (4.401) | 31.231 (71.743) | 0.0058 (0.006) | 0.497 (0.499) |
| VIMPNN | Augmented QM9 | **0.065** (0.197) | **0.043** (1.093) | **0.0008** (0.002) | **0.033** (0.047) |
| | Infinite Crystal | 0.034 (0.044) | 0.003 (0.009) | 0.0015 (0.002) | **0.048** (0.035) |
| | Stable CG | **0.513** (0.537) | **0.552** (1.162) | **0.0016** (0.002) | - |
| | Unstable CG | **2.387** (3.557) | **18.336** (56.050) | **0.0042** (0.004) | **0.162** (0.237) |

observation that the bond type is a good feature for the estimation of energy. Furthermore, the use of bond type information in the design of the NN architecture (case a.ii in Section 4.2) has a stronger positive impact than using it as a feature in MPNN. This suggests that having specialised messaging channels based on bond type better captures the physics of atomic interactions than just including bond type information in generic messages. However, MPNN does improve significantly on the baseline, therefore in future works it may be interesting to combine these two strategies.

The auxiliary estimation of physical properties also improve on the baseline model, but less so than the integration of bond type information. We interpret this as a result of the model having to implicitly discover and encode the useful physics representations required for accurate predictions. Indeed, the auxiliary quantities used are related to the global properties of the molecule, and in the case of the distance scaling to a high level system optimisation. In future works, it may be interesting to investigate the effects of auxiliary quantities linked to the physics of individual atoms/nodes. The more modest effect of auxiliary properties may still be beneficial in combination to the bond type strategy, as we demonstrate in the next experiment and the Appendix, thus we continue using them.

## 5.2 EVALUATION OF VIMPNN ON OUT-OF-EQUILIBRIUM MOLECULAR AND CRYSTAL DATA

In this experiment, we update the baseline model of Section 5.1 with our physics integration strategies, namely bond type-specialised communication channels and node updates (case a.ii of Section 4.2) and all 3 of the auxiliary estimations to create VIMPNN. The energy estimation is assessed on the different scenarios provided by our 3 datasets. On the Crystal Growth subsets, we consider system sizes ranging from 15 to 75 atoms, instead of the available 114, due to memory restrictions. The performance of MPNN and VIMPNN are reported in Table 2 using the same metrics as previously, and the distance to stable configuration (DSC) where the interatomic distance at the minimum of the predicted energies is compared to the known at-equilibrium distance by absolute difference.

Our first observation, when comparing with Table 1, is that the auxiliary estimates, which were found to be beneficial to the model in the previous experiment when used in isolation, also prove to be beneficial when used in conjunction with the bond type-specialised communication channels.

VIMPNN generally obtains better scores than MPNN on all datasets, meaning that energy estimates are similar or more accurate and that their minimum is closer to the true stable interatomic distance (Fig. 3 left). The improvement over MPNN is particularly strong on the Crystal Growth dataset, hinting that VIMPNN is able to generalise better to new geometries. We also note that MPNN's metrics, especially DSC, suffer from a high standard deviation. Upon examination, we find that

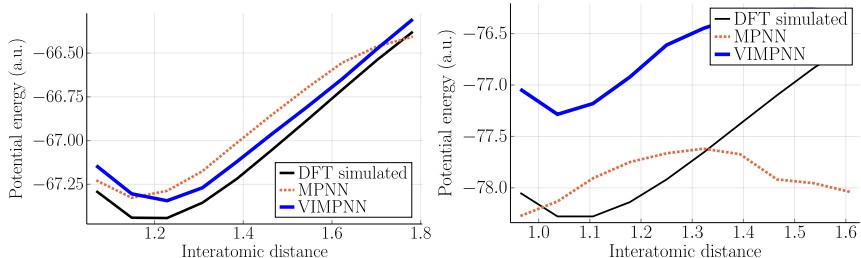

Figure 3: Energy estimations at different interatomic distances for MPNN and VIMPNN. Two examples cases are shown for the C3H4O3 and C5H7N3O molecules in the Augmented QM9 dataset.

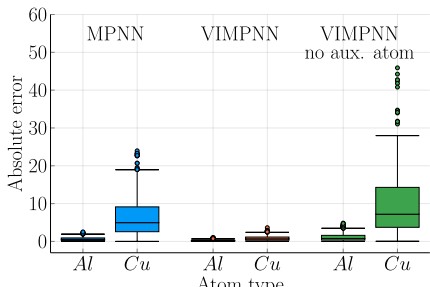

Figure 4: Evaluation of MPNN and VMPNN's ability to handle different atom types.

there exists some more difficult cases[4] which may explain these results. Indeed, when examining these cases more closely (Fig. 3 right), we find that MPNN's estimations are usually close to the true energy values but do not contain a minimum. On the other hand, while VIMPNN's estimates are slightly worsened by an offset, it is still able to account for the physics of the system, such as the repulsive effect of atoms getting too close to each other, to predict an energy minimum at a reasonable interatomic distance. The energy offset may be addressed in future works, for example by a different loss term or new auxiliary estimates. The Infinite Crystal dataset is the only case where VIMPNN doesn't improve on MPNN in term of the accuracy of energy estimation, both methods obtaining similar scores, but with VIMPNN still producing better DSC measures by almost an order of magnitude. As we only consider two mono-atomic infinite crystals in our experiments, we may see further improvements by expanding this dataset to include a higher number of structures outside of the fcc lattice and a wider variety of atom types.

Furthermore, when trained on the Crystal Growth dataset, the results of MPNN on Cu crystals are noticeably worse than for Al (see Fig. 4). We interpret this as MPNN not being able to learn how to handle both crystal types simultaneously. VIMPNN, however, maintains a similarly good accuracy in both cases. We hypothesise that the auxiliary estimations of number of each type of atom may help VIMPNN to better differentiate between these atom types. We test this hypothesis by removing the atom-type estimation from VIMPNN. The resulting high errors for Cu crystals demonstrate that the auxiliary estimation has been an effective way to integrate knowledge into the model, in this case by directing the attention of the model to atom type, a very relevant physical property for this problem. Other combinations of auxiliary estimations have been tested in the Appendix (Table 4).

We further evaluate how VIMPNN scales to larger systems by comparing energy estimates on crystalline systems of various sizes (i.e. number of atoms) from the Stable Crystal Growth subset. The increasing energy errors for larger sizes in Fig. 6 left suggest that the model struggles with handling many simultaneous interactions at a higher range of interatomic separations. When examining examples of such systems more closely, e.g. in Fig. 5, we observe that VIMPNN finds an energy minimum at the correct stable interatomic distance for the small system (Fig. 5 left, 15 atoms) even if the estimated energy has an offset, but in extreme cases (Fig. 5 right, 75 atoms) the predicted energies

---

[4]The cause to these more difficult cases is still unknown and will be investigated in future work.

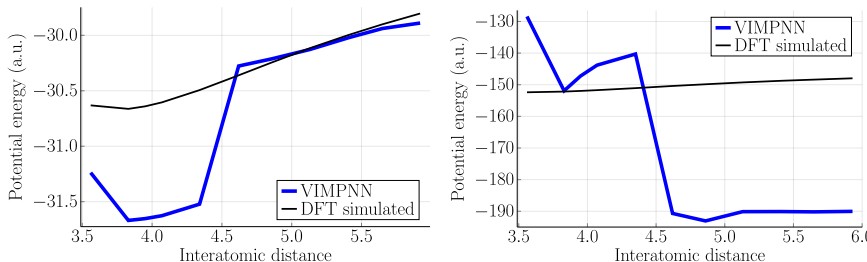

Figure 5: Energy estimations for a small (15 atoms, left) and large (75 atoms, right) crystalline system.

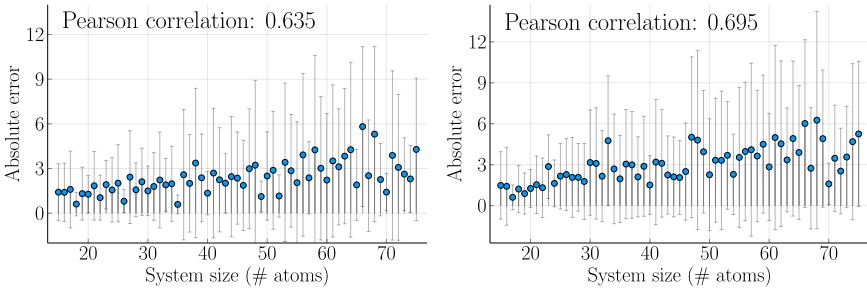

Figure 6: MAE of energy estimations w.r.t system size from the Stable Crystal Growth dataset. VIMPNN is trained on (left) all system sizes of 15-75 atoms, and (right) systems of up to 25 atoms.

no longer present a correctly located minimum. These results indicate that further work is needed towards better accounting for atomic interactions within large scale structures.

### 5.3 GENERALISATION TO LARGER SYSTEMS

In this section we investigate VIMPNN's ability to learn basic principles of atomic interactions in small chemical systems that are transferable to arbitrary (especially larger) system sizes. This ability would allow for quicker training times while generalising well irrespective of the graph size. To test whether VIMPNN is capable of such task, we train the model on a filtered version of the Stable Crystal Growth subset that includes only systems of up to 25 atoms. The validation and testing are done on the full subset, thus the model estimates energies for systems of up to 75 atoms[5]. Fig. 6 (right) shows that the MAE increases with the system size above 25 atoms, but this trend is only marginally stronger than when the model is trained on all sizes (Fig. 6 left, of the previous experiment). This indicates that VIMPNN is learning some basic principles about the atomic interactions that are transferable to larger, unfamiliar, systems. However, for VIMPNN to make best use of this positive effect, future work will need to address the overall performance on large system sizes.

### 5.4 INTERPRETATION OF THE NODES' HIDDEN STATES

When using VIMPNN in-place of DFT, we are placing trust in the model for computational speed with minimal impact in accuracy. While DFT includes approximations to make computations more tractable, it is grounded in fundamental principles of atomic interaction. VIMPNN however makes inferences about the system's energy by searching for a generalised function that matches well the input (chemical graph) and output (energy) data. It is therefore helpful to investigate the learnt representations of atomic interactions within the NN, to understand how to interpret and trust the model's estimates and its ability to generalise to new systems. To do so, we visualise the hidden states of the nodes during the readout phase (step 3 of Section 4.1) that conveniently reduces the nodes' vector to scalar values before summing to produce the energy estimate. Thus, these reduced

---

[5]As in Section 5.2, larger systems up to 114 atoms are not used due to memory constraints.

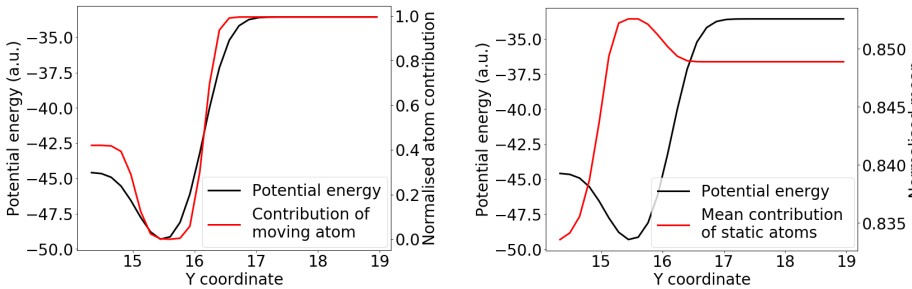

Figure 7: Contributions of a moving atom (left) and static crystal (right) toward the energy estimate.

node values may be considered as the contributions of the nodes to the energy estimate, and these values, normalised across the nodes, are referred to as *node's contribution* in the rest of this section.

After training VIMPNN on the Unstable Crystal Growth subset, we consider the node of an atom that is newly added to a (stable) crystal seed of 14 atoms. When compressing and dilating the distance of this (single) atom to the rest of the crystal, we examine its node contribution and that of the rest of the (static) atoms (Fig. 7)[6]. At its stable location, the atom's contribution is marginal, in line with the crystal being at its minimum energy driven by the regular structure of the crystal. As the atom is moved closer or pulled away, its contribution increases simultaneously with the potential energy. At the same time, the relative contributions of the static atoms slightly decrease to give way to the perturbation of the moving atom. This strongly suggests that the model learnt to pay attention to the location of individual atoms, although it was trained on systems that are isometrically compressed/dilated. This confirms the previous experiment's conclusion that VIMPNN learns transferable principles of atomic interaction.

This visualisation method could inform further improvements in the model, such as related to the aforementioned issues of large range atomic interactions. It may also be helpful in the simulation of crystal growth where a lattice is not imposed, by pinpointing stable atomic positions that entail equal contributions of all atoms to the system's energy.

## 6 CONCLUSION

We have shown how integrating physics knowledge into data driven learning models may help improve accuracy and generalisation. We proposed two integration strategies, namely specialised information passing channels within a DNN to better account for bond type in the simulation of atomic interactions, and estimating auxiliary physical properties that further relate the learnt representations to the underlying physics of the studied phenomenon. Our methods are incorporated into a Valence Interaction Message Passing Neural Network (VIMPNN) that estimates the potential energy of both molecular and crystalline systems. The energy estimate is shown to be fine enough to identify stable spatial configurations that correspond to an energy minimum. Three datasets are introduced to benchmark the performance of DNN models: an augmented QM9 dataset to include new unstable configurations of molecules, infinite-size periodic crystals, and a more complex stable and unstable Crystal Growth dataset with crystalline systems at varying sizes and different spatial configurations. We evaluate both VIMPNN and MPNN on these datasets, and find that our model performs as well or better than MPNN. VIMPNN proves able to learn principles of atomic interactions that are transferable to new system sizes and spatial configurations. However, more work is needed to attain a more robust generalisation to larger chemical systems. This may be achieved in future works by integrating more physics knowledge on long-range atomic interactions into the model.

---

[6]Note that this scenario is different to previous experiments with compressing/dilating the whole system. Still, VIMPNN produces very plausible energy estimates in Fig. 7.

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

# A APPENDIX

## A.1 FURTHER ASSESSMENT OF THE PROPOSED PHYSICS INTEGRATION STRATEGIES

Table 3: Comparison of the different implementations of bond type specialised communication channels (see Section 4.2), tested on the Augmented QM9 dataset.

| Implementation | MAE | MSE | RE |
|---|---|---|---|
| a.i | 0.1214 (0.2952) | 0.1019 (1.1220) | 0.0016 (0.0036) |
| a.ii | **0.0670** (0.1408) | 0.0243 (0.4059) | **0.0009** (0.0016) |
| b.i | 0.1816 (0.5816) | 0.3712 (3.8593) | 0.0023 (0.0061) |
| b.ii | 0.1079 (0.1206) | **0.0223** (1.1093) | 0.0012 (0.0015) |
| c | 0.1144 (0.1418) | 0.0332 (0.0920) | 0.0015 (0.0019) |

Table 4: Evaluation of the auxiliary estimations combined with the selected bond type integration strategy (a.ii), on the Augmented QM9 dataset.

| Auxiliary combination | MAE | MSE | RE |
|---|---|---|---|
| BT + # atoms | 0.0690 (0.1921) | 0.0409 (0.9880) | 0.0009 (0.0021) |
| BT + # orbitals | 0.0796 (0.1357) | 0.0220 (0.4670) | 0.0009 (0.0015) |
| BT + distance scaling | 0.0742 (0.1450) | 0.0265 (0.4647) | 0.0010 (0.0017) |

## A.2 THE EFFECT OF VARYING TIMESTEPS

In all experiments of this paper, three iterations of information passing between nodes were used as the basis for both VIMPNN and MPNN following the default recommendation from Gilmer et al. (2017). However, we also experimented with both 1 and 8 timesteps, with results reported in Table 5. We found empirically that 3 iterations works best for VIMPNN as well.

Table 5: Performance on the Augmented QM9 dataset with varying number of timesteps $T$.

| Model | # timesteps | MAE | MSE | RE |
|---|---|---|---|---|
| | 1 | 0.1860 (0.338) | 0.1488 (1.341) | 0.0026 (0.004) |
| VIMPNN | 3 | **0.0646** (0.197) | **0.0430** (1.093) | **0.0008** (0.002) |
| | 8 | 0.0944 (0.124) | 0.0243 (0.067) | 0.0013 (0.001) |
| | 1 | 0.1737 (0.298) | 0.1191 (1.438) | 0.0024 (0.003) |
| MPNN | 3 | **0.0909** (0.476) | **0.2348** (6.679) | **0.0012** (0.005) |
| | 8 | 0.5550 (0.352) | 0.4321 (0.496) | 0.0077 (0.005) |

## A.3 IMPLEMENTATION DETAILS

All models were trained using an Nvidia Tesla V-100 with PyTorch 1.2.0, Python 3.7.2, and CUDA 10.1. Each model was trained for a maximum of 360 epochs, with 50 epochs of no validation loss improvement as the criteria for early stopping. During training, the best model (determined by the validation loss) was saved to be used for testing.

The following hyperparameters were used for all experiments:

- Batch size: 20 for the Augmented QM9 and Infinite Crystal datasets, and 5 for the Crystal Growth dataset due to memory constraints.
- Learning rate: 1e-4
- Learning rate decay: 0.6
- Early stopping: 50 epochs
- Auxiliary alpha: 0.3

- Size of bond type specialised message $m_v|_{BT}$: 73
- Number of bond types: 3
- Number of message passing iterations: 3

When training with the different implementations of the bond type integration, though the size of the bond type specialised message $m_v|_{BT}$ remains consistent at the aforementioned 73 dimensions, in case b) the concatenation of bond type messages entails that the nodes' hidden state size is 219. For clarity, we detail the size of nodes' hidden state here for each of the methods:

- a: 73
- b: 219
- c: 73

