# OpenReview forum: "VIMPNN: A physics informed neural network for estimating potential energies of out-of-equilibrium systems"
_ICLR.cc/2020/Conference — Reject_

### Official Review · AnonReviewer1 · 2019-10-13
**Official Blind Review #1**

**Rating:** 6

**Review:**

This paper presents a number of new / extended datasets for the evaluation of ML-based prediction of energies of unstable systems, as well as a network (VIMPNN) that includes a new and better way of including bond-type information. It is also proposed to use auxiliary losses (predicting other chemical properties).

Although I am not an expert in chemistry, the new datasets seem fairly well thought out and their utility is well motivated. The proposed change to the MPNN network architecture is rather simple and hardly physics inspired, but the empirical improvement seems substantial, so this too is a nice contribution. So I have decided to give the “weak accept” rating.

In 4.2 it is explained how different ways of incorporating bond information were evaluated, and it is stated that “best results were obtained in the case a.ii). However, no results are presented to support this claim, leaving the reader to wonder how rigorous this exploration was. I would suggest systematically evaluating the different options and including the results in an appendix.

In table 1 results are shown for the MPNN baseline, baseline with specialised node updates a.ii, and with auxiliary estimates. However, combinations of these are not evaluated. Nevertheless, if I understood correctly, the VIMPNN method tested later includes all of the separate improvements. It would be good to include experimental results to motivate this.

As acknowledged in the paper, the idea of using bond-type information was already in Gilmer et al. Also, I think the different ways of including bond-type explored in this paper are not really informed by physics. The choice for method a.ii is made based on empirical results. This is not a problem in itself, but I would suggest that the authors change the wording to not over-promise on the physics-inspiredness. E.g. the abstract says “VIMPNN integrates prior knowledge such as the existence of different interatomic bonds”, suggesting that there is more prior knowledge being exploited than just bonds.


Comments:

“It produces comparable accuracy to that of DFT while also improving computation time by 5 orders of magnitude”.
I assume this speedup is relative to DFT. It would be good to be explicit about that, and also discuss the speed relative to the MPNN baseline (I suppose MPNN and VIMPNN are similar).

“The change of atomic distances are performed isomorphically” - I would say “isometrically”.


**Experience Assessment:**

I have read many papers in this area.

**Review Assessment: Checking Correctness Of Derivations And Theory:**

I carefully checked the derivations and theory.

**Review Assessment: Checking Correctness Of Experiments:**

I assessed the sensibility of the experiments.

**Review Assessment: Thoroughness In Paper Reading:**

I read the paper at least twice and used my best judgement in assessing the paper.

---

> ### Author Response · Authors · 2019-11-13
> **Response to Review #1**
>
> We thank the reviewer for their comments. We have responded to the feedback:
>
> "In 4.2 it is explained how different ways of incorporating [...] I would suggest systematically evaluating the different options and including the results in an appendix. "
>
> We agree that the paper would benefit from adding the exploration of different bond type integration strategies into an appendix. We have added these values in Table 3.
>
> "In table 1 results are shown for the MPNN baseline [...] It would be good to include experimental results to motivate this."
>
> The purpose of Table 1 was to evaluate each proposed knowledge integration strategy in turn, against the baseline of no integration at all, and against MPNN. We have now made this clearer in the text of section 5.1. It is true that VIMPNN, in section 5.2 and following, includes all these improvements. We agree that some intermediate evaluations with combinations of some improvements would provide a fuller analysis of their effect in combination. We have now added additional results in Table 4.
>
> In the sake of time, given that the bond-type integration is the most effective improvement, we’ve elected to experiment with various combinations of bond-type plus-auxiliary instead of all possible combinations.
>
> "As acknowledged in the paper, the idea of using bond-type information was already in Gilmer et al. [...] suggesting that there is more prior knowledge being exploited than just bonds."
>
> As we demonstrate in Table 1, our new strategy for integrating bond type information is more effective than the one in Gilmer et al. . This strategy is to make the information transfer within the NN closer to the physics of atomic interaction through using bond type specialised communication channels, while Gilmer et al. were only using bound type information as a feature. Our principle is directly inspired by the physics of atomic interactions, as we now explain in more detail in sections 4.2 and 5.1.
> However, it may be implemented in various ways, and we experimented with several as the reviewer rightly highlighted, which indeed are driven by experiments rather than physics.
>
> In addition, more physics knowledge is exploited thanks to our second knowledge integration strategy. Indeed, we present the estimation of auxiliary properties as a way to integrate prior knowledge into the model. Indeed, the choice of auxiliary properties allows directing the attention of the model to properties of interest within the problem — in the present case the type of atoms, as highlighted by the newly added experiment in Section 5.2, some of their physical properties that are relevant to the problem, or higher level properties such as the localisation of stable atom positions. The reviewer’s comment indicates that this needs to be clarified in the text, which we now do in Section 5.2. More experiments will be performed in future work to further explore and evaluate this knowledge integration strategy.
>
> As per the reviewer’s suggestion, we modify the abstract to better explicit the way physics knowledge is exploited into our model: VIMPNN integrates prior physics knowledge, namely the information exchange between atoms being driven and modulated by different interatomic bonds, and the relevance of specific physical properties to the problem of estimating more accurate energies.
>
> “It produces comparable accuracy to that of DFT [...] It would be good to be explicit about that, and also discuss the speed relative to the MPNN baseline"
>
> Indeed, that is correct. We have added the clarification for both the speed w.r.t DFT and MPNN in section 1: It produces comparable accuracy to that of DFT while also improving the computation time by 5 orders of magnitude. We demonstrate that our method also produces more accurate energy estimations than that of MPNN, while being 30% slower at 0.007 seconds per energy estimate, which is marginal compared to the gain from DFT and may be improved with some obvious code optimisation.
>
> "“The change of atomic distances are performed isomorphically” - I would say “isometrically”. "
>
> Thank you for pointing this out, we’ve changed it to “isometrically” throughout the article.

---

### Official Review · AnonReviewer3 · 2019-10-20
**Official Blind Review #3**

**Rating:** 6

**Review:**

The paper tackles the problem of estimating the electronic structure's ground state energy of a given atomic system by means of supervised machine learning, as a fast alternative to conventional explicit methods (DFT).
This is done by improving on a previous method, MPNN, Message Passing Neural Networks, and in particular by including information on bond type as input, so that the NN can learn the appropriate weight for messages going through bonds of that type.  In addition, training on several target labels (multi-regression) is attempted, with the idea that more physical outputs may help building better hidden representations (on this, there are mixed results).  Separately, training sets are enriched with non-equilibrium structures, so as to confront the NN with more diverse data.

The method is tested on 3 kind of training sets. The first is a simple but time-consuming augmentation of QM9, with inter-atomic distances varied, so as to increase the training set's size (and in particular, including non equilibrium configurations). The second consists in a periodic and thus infinite simple crystal structure (with, again, variations in inter-atomic distances, enriching the dataset). The third is a pseudo-cristalline structure with atoms randomly placed on a regular grid, forming a somewhat random structure, also named crystal (this is not a very good name).

The paper is overall rather well written, sometimes being a bit cumbersome (long sentences), but mostly it is stating clearly what is done or discovered.  The work is situated within the existing literature (that I am not familiar with at all).  The idea of using physics to guide architecture choices is gaining a lot of attention recently and seems to be well-suited to this particular problem, and well applied.  Several ways to use the bond type information have been attempted in this work, and several of them are reported and compared (a couple of them are discarded).  The results convincingly show that using bond type information indeed increases performance, both for small systems and for regular crystals.  The impact of performing multi-regression is less important, but still positive.  For large ''random crystals'', the method does not perform very well, and this represents a challenge for future work. Such a confession on the method's limitations is welcomed.

Given the idea (using physical information as bond type) is clearly and honestly presented, produces significant improvement compared with previous works, and has perspective for multiple future developments, I recommend acceptation of this paper.


There are however a number of points that could be improved.

1. There is a physical mistake that is not crucial but should be corrected, when talking about the ground state, and in particular before this sentence: ''accurate ground-state energy estimation of out-of-equilibrium molecule''. Ground state means minimal, T=0K energy level, so by definition it is at equilibrium. Thus, the sentence seems quite contradictory to a physicist.
What DFT and VIMPNN actually compute is the electronic structure's ground state's energy (at fixed positions of the atom kernels). I think this distinction should be mentioned just once, and then you could proceed with saying ground state energy.
Because of this, I would recommend to edit the title so as to suppress ''out of equilibrium'' from it.  Otherwise readers may think the method deals with non-equilibrium electronic structures (non ground states), which clearly it does not at all, or they may think that it is especially good at estimating energies for out of equilibrium systems, which is not its primary goal.

2. I do not understand very well the training procedure.  Also there are some tests that seem to be interesting and that are not performed (as far as I understood).
Does each training set contains the 90%-150% data augmenations, for each non-augmented training configuration ?
Why is training performed separately for each kind of data set ? Wouldn't the ultimate goal be to transfer learning from a type to another, e.g. from QM9-style to crystal style, etc ?  (as far as I understood, this was not done)
Isn't it interesting to see how much training on an augmented data set (let's say QM9) improves performance on the non-augmented data (the ''true data'' in a sense) ? Although the augmented data is obtained by DFT, and comparing models trained on different data sets is unfair, I think it may be interesting to see if VIMPNN benefits more than MPNN from this strategy (so compare the performance gains of both algorithms obtained by augmenting a data set). This kind of comparison may also be done for the ''augmentation'' of a training set by the concatenation of it with another one (although in that case it may be detrimental to the test accuracy?).
If you actually did some of this, then I misunderstood and I am sorry, but then this also means you should clarify.

3. Section 3.3:
I would not call this a crystal, but more something like ''random finite structure''. This should be done everywhere in the paper.

4. Section 4.1 is a bit too short for the inexperienced reader. I suggest to be a bit more explicit on what is learned

5. Section 4.2: It is nice to say you tried other ways, keep that.
However, try to be more explicit on what is shared and what isn't, in the architecture you finally pick. Is lambda(v,w) a common value for all bonds of the same type, like C-C ? Maybe you could provide an example or some more detailed notation to make your choice fully explicit (after all this is the core of the paper).

6. Section 4.3: could you quickly comment on why you don't use more of the 13 physical observables available in QM9 ?

7. Table 1: do the three last lines correspond to ''no BT information + a single auxiliary estimate'' ?  It seems to be the case, but then you say you will continue with the auxiliaries, in addition to the BT information.  Why don't you display the result of using BT information AND the 3 auxiliary estimates ?  If you did, then I misunderstood, but it would also mean you did not explain well enough a counter-intuitive result (which would be that adding auxiliary information actually hurts the performance of the VIMPNN).

8. Please include a couple of explicit numbers of your training/test/validation sets sizes.

9. section 5.2 could be made more concise. In particular, there is no need in repeating what can be seen directly in the figures (stating numbers). It is useful to comment on the meaning of the results however (as you currently do).

10. section 5.4  is a good idea, promising, however it does not really conclude into a very strong statement, and takes essentially 1 full page, which could be used to better clarify the architecture and/or the training procedure (or to reduce towards the ideal page length).





**Experience Assessment:**

I do not know much about this area.

**Review Assessment: Checking Correctness Of Derivations And Theory:**

N/A

**Review Assessment: Checking Correctness Of Experiments:**

I assessed the sensibility of the experiments.

**Review Assessment: Thoroughness In Paper Reading:**

I read the paper at least twice and used my best judgement in assessing the paper.

---

> ### Author Response · Authors · 2019-11-13
> **Response to Review #3**
>
> We would like to thank the reviewer for carefully evaluating our paper and for his/her constructive suggestions. Our responses are as follows:
>
> 1. There may be a confusion coming from what is defined or meant by "ground state" and "equilibrium", i.e. whether it concerns the structural ground state (positions of atoms) or the electronic ground state. We thank the reviewer for pointing this out, and we clarify this in the revised version of our article. We always compute the electronic ground state, therefore we have made this clear in section 1.
>
> 2. Yes. The training data consists of all data augmentations for 80% of the configurations. This means, that the training data for the molecule dataset is 8000 different molecules, each with 10 variations of interatomic distance. VIMPNN learns from examples of (un)stable configurations. Then, it performs inference on unseen systems with no prior knowledge of the stability of the configuration. We make this more clear in section 5.
>
> "Why is training performed separately [...] from QM9-style to crystal style, etc ?"
>
> Training was done on the different datasets to demonstrate the versatility of VIMPNN on different case studies, as we now highlight in section 5. Though, we do agree that transfer learning would be a very interesting experiment as it may improve VIMPNN’s generalisation. We thank the reviewer for this suggestion and will include it in future works, as there may not be enough time to include it in the current article.
>
> "Isn't it interesting to see how much training on an augmented data set [...]"
>
> We like the idea of comparing the performance with the augmented dataset and on the original dataset, respectively. It is indeed possible that the augmentation improves on the estimation accuracy. However, we fear that the comparison may not be fair, as one model would be trained on 10 times more data than the other.
>
> "This kind of comparison [...] it may be detrimental to the test accuracy?)."
>
> As with point 2b), we have not combined the datasets so far, but in future experiments it may indeed be interesting to do so, to investigate the generalisation ability of the model.
>
> 3. While strictly speaking crystals are periodic in nature, what we are simulating in this dataset is crystal growth. When creating our dataset we use a crystal structure as a start point, then add atoms one by one on the surface of the growing crystal following the lattice pattern. Even if some atoms are missing in the pattern and the systems are not fully periodic, they are representative of a growing crystal. We make this clearer in sections 3.3. Would the reviewer consider ‘crystal growth dataset’ to be a more suitable name for the dataset and ‘crystalline systems’ for its elements?
>
> 4. We have edited section 4.1 to provide more details on the different components of MPNN.
>
> 5. Yes, the bond type coefficient has the same value for all bonds of the same type. The notation was indeed badly chosen and we have modified it in the revised version of the paper as lambda_BT. We also provided the equations for the other approaches where we felt it would disambiguate the explanations, with the exception of approach b ii) which would have required detailing the GRU equations, which would take too much space and may be out of the scope of this paper.
>
> 6. With MPNN, the best performance for the 13 physical properties were obtained through training a separate MPNN on each property. Although we can also train one VIMPNN per property, we decided to focus on the ground-state energy as a proof of concept. This property is of particular interest for us because it allows finding stable interatomic distances, which is relevant to our augmented datasets and to simulating crystal growth. We may experiment with additional physical properties in future works, either with specialised VIMPNN models or in combination within a unique VIMPNN model.
>
> 7. In Table 1, the last three lines indeed correspond to “no BT information + a single auxiliary estimate”. The aim of this table was to evaluate the impact of each augmentation of the model, so in isolation. The results of using BT information AND the 3 auxiliary estimations are provided in Table 2. This is what we denote as VIMPNN architecture. As shown in Table 1, adding the auxiliary estimations has a small positive impact on the performance. Table 2 shows that this small impact is also present when BT information is used. We now clarify and comment on these in Sections 5.1, 5.2.
>
> 8. We have added the splits sizes for the training, testing, and validation sets into the second paragraph of section 5.
>
> 9. We have made section 5.2 more concise as suggested, but we also added some additional experiments following the suggestions of other reviewers.
>
> 10. Thank you for the suggestion. We have updated section 5.4 to be more concise and accurate in its description, and we improved it with a concluding statement on the current interpretation and future use of this visualisation.

---

### Official Review · AnonReviewer4 · 2019-11-10
**Official Blind Review #4**

**Rating:** 6

**Review:**

This paper studies approximation of the potential energy of molecules by a message passing architecture. The work builds upon [Gilmer et al., 2017] and the contributions are two-fold:
1) The creation of new datasets to learn and test such architectures on and the augmentation of an existing dataset in order to account for energies at non-equilibrium states.
2) A proposed modification to the MPNN architecture proposed in [Gilmer et al., 2017], in order to account for physical properties in the message-passing procedure.
The performances of the architectures are studied with numerous numerical experiments.
The paper is overall well-written and clear.

The new dataset utility is sound and well-motivated. Unfortunately I can not further motivate upon this, as I am not familiar with this area.

From the point of view of the proposed architecture, the work is quite incremental. The bond type information, previously included as feature, is now transferred to an architectural modification. On the other hand, many different modifications (although no substantially different from each other) are proposed and tested (although no results about the different modifications are reported - it could be nice to have them in an appendix). This motivates the ‘weakly accepted’.

The authors also considered the idea of adding additional learning modules (and a related loss) to help the model learn more 'physics interpretable' hidden states. While it does not seem to give notable gains here, it is an interesting idea and I believe deserves further experimentations in the future.

The experiments are numerous and various, and they offer a very good overview on the goodness of the model (and its limitations). They first compare with the baseline on the augmented dataset, and they show notable gains on the MPNN baseline. The ability of the network to reproduce the energy curve at different interatomic distances is then studied on the different dataset and in different settings, showing gains over the baseline. The authors also report some negative results and experimental interpretation of the model hidden states, which are also an important contribution in my opinion.

Further comments:

1. No details are given about the training of the models. I think a small paragraph (or larger and reported in the appendix) should be added.

2. Even if it builds upon previous work, the (VI)MPNN model may be further explained. For example, what type of functions are M_t and R? The explanation on the considered modifications of MPNN may be clearer (maybe with the introduction of a more mathematical notation).

3. How long is the message diffusion (T)? What is the effect of larger / smaller T’s?

4. What’s the point of equations (5) (6) (7)? They are exactly the same and they do not add any information. It would be more useful to explain what type of function R is in my opinion.

5. Table 1, Auxiliary estimates: Are these the results obtained by the model a.ii trained to jointly learn the energy and the properties i) ii) iii) ? In what sense they improve on the baseline? This part was not clear to me.

6. Section 5.2: ‘[…] we combine our proposed physics integration strategies, namely bond type
specialised
node updates (case a.ii) of Section 4.2) and auxiliary estimations of physical properties,
into the VIMPNN model […]’. I do not understand this sentence. Isn’t in fact the VIMPNN architecture the same as MPNN with the modification a.ii? In what sense do you integrate a.ii in it?

7. If I understood correctly, the main final objective is to be able to characterize the minima of the energy. In this case, could you asses the performances of the two methods (MPNN) and (VIMPNN) by measuring some kind of distance from the approximated minima from the actual one?

Typos:

Section 5.3 first line: ‘We investigate the VIMPNN’s the ability …’ -> ‘We investigate the VIMPNN’s ability’

Section 5.2 Fourth sentence: ‘The first seeks to demonstrates a the model’s …’ -> ‘The first seeks to demonstrates the model’s…’

**Experience Assessment:**

I do not know much about this area.

**Review Assessment: Checking Correctness Of Derivations And Theory:**

N/A

**Review Assessment: Checking Correctness Of Experiments:**

I assessed the sensibility of the experiments.

**Review Assessment: Thoroughness In Paper Reading:**

I read the paper at least twice and used my best judgement in assessing the paper.

---

> ### Author Response · Authors · 2019-11-13
> **Response to Review #4**
>
> We appreciate the detailed feedback given by the reviewer. We have prepared a response:
>
> "many different modifications (although no substantially different from each other) are proposed and tested (although no results about the different modifications are reported - it could be nice to have them in an appendix)."
>
> We have reported the additional results in Table 3 of the Appendix.
>
> "The authors also considered the idea of adding additional learning modules (and a related loss) to help the model learn more 'physics interpretable' hidden states. While it does not seem to give notable gains here, it is an interesting idea and I believe deserves further experimentations in the future."
>
> We thank the reviewer for his encouragements. We have added an experiment in Section 5.2 to examine the benefits of one of these additional learning modules in learning to handle two types of crystals (Al and Cu) simultaneously. We also added some evaluations of performance when combining these modules to the bound type driven architecture in the Appendix. Further experimentations will be done in future works.
>
> "1. No details are given about the training of the models. I think a small paragraph (or larger and reported in the appendix) should be added."
>
> We have improved the description of the training procedure in Section 5, and added a section on hardware and training hyperparameters to the appendix.
>
> "2. Even if it builds upon previous work, the (VI)MPNN model may be further explained. For example, what type of functions are M_t and R? The explanation on the considered modifications of MPNN may be clearer (maybe with the introduction of a more mathematical notation). "
>
> MPNN has been explained in more detail in Section 4.1, especially the purpose and functioning of M_t and R. We also improved the description of VIMPNN in Section 4.2.
>
> "3. How long is the message diffusion (T)? What is the effect of larger / smaller T’s?"
>
> For all our experiments we use 3 diffusion steps which is the default value of MPNN. We clarify this in the new ‘Implementation details’ section of the Appendix. We followed the recommendations of Gilmer et al. and tested with 1 to 8 timesteps, and found empirically that 3 iterations works best for VIMPNN as well.
>
> The question of the effect of smaller/larger T is definitely interesting, and we have started investigating it, as can be seen in the new Table 5 in the appendix. However, we feel that more experiments are needed to answer it satisfactorily, therefore this will be addressed in a future paper.
>
> "4. What’s the point of equations (5) (6) (7)? They are exactly the same and they do not add any information[...]"
>
> We agree that these equations are not very informative, and we have removed them. We added some pointers as to how M_t and R are implemented instead, i.e. using neural networks.
>
> "5. Table 1, Auxiliary estimates: Are these the results obtained by the model a.ii trained to jointly learn the energy and the properties i) ii) iii) ? In what sense they improve on the baseline?."
>
> The results in table 1 show the performance of each of the physics integration methods taken in isolation. They are only combined in Section 5.2 and later. We have now clarified this in Section 5.1. We also added results of each auxiliary estimate combined with model a.ii in Table 4 of the appendix.
>
> "6. [...] Isn’t in fact the VIMPNN architecture the same as MPNN with the modification a.ii? In what sense do you integrate a.ii in it?"
>
> MPNN (with no bound type information, so the baseline in Section 5.1) is indeed the base model which we improve by adding our proposed physics integration strategies, namely BT specialised communication channels and auxiliary estimations. We rephrase this sentence to better reflect this: we update the baseline model of Section 5.1 with our proposed physics integration strategies, namely bond type-specialised communication channels and node updates (case a.ii) of Section 4.2 and all 3 of the auxiliary estimations to create VIMPNN.
>
> "7. If I understood correctly, the main final objective is to be able to characterize the minima of the energy. In this case, could you assess the performances of the two methods (MPNN) and (VIMPNN) by measuring some kind of distance from the approximated minima from the actual one?"
>
> Thank you for the suggestion, we have included this to table 2 to show the mean absolute distance from the true minima and the estimated one.
>
> "Typos:
> Section 5.3 first line: ‘We investigate the VIMPNN’s the ability …’ -> ‘We investigate the VIMPNN’s ability’
> Section 5.2 Fourth sentence: ‘The first seeks to demonstrates a the model’s …’ -> ‘The first seeks to demonstrates the model’s…’"
>
> Thank you for pointing out these errors. They have been amended in the revised copy.

---

### Decision · Program_Chairs · 2019-12-19

**Decision:**

Reject

**Comment:**

The paper considers the problem of estimating the electronic structure's ground state energy of a given atomic system by means of supervised machine learning, as a fast alternative to conventional explicit methods (DFT). For this purpose, it modifies the neural message-passing architecture to account for further physical properties, and it extends the empirical validation to also include unstable molecules.

Reviewers acknowledged the valuable experimental setup of this work and the significance of the results in the application domain, but were generally skeptical about the novelty of the machine learning model under study. Ultimately, and given that the main focus of this conference is on Machine Learning methodology, this AC believes this work could be more suitable in a more specialized venue in computational/quantum chemistry.